# Record Linkage of Chinese Patent Inventors and Authors of Scientific Articles

Robert Nowak [1,*], Wiktor Franus [1], Jiarui Zhang [2], Yue Zhu [2], Xin Tian [2], Zhouxian Zhang [2], Xu Chen [2] and Xiaoyu Liu [2]

1   Institute of Computer Science, Warsaw University of Technology, 00-665 Warsaw, Poland; wiktor.franus@mion.pw.edu.pl
2   Shanghai Science and Technology Development Co. Ltd., Shanghai 200233, China; jrzhang@sstir.cn (J.Z.); yzhu@sgst.cn (Y.Z.); xtian@sstir.cn (X.T.); Cho_Z@outlook.com (Z.Z.); xchen@sstir.cn (X.C.); yjhu@sstir.cn (X.L.)
*   Correspondence: robert.nowak@pw.edu.pl

**Abstract:** We present an algorithm to find corresponding authors of patents and scientific articles. The authors are given as records in Scopus and the Chinese Patents Database. This issue is known as the record linkage problem, defined as finding and linking individual records from separate databases that refer to the same real-world entity. The presented solution is based on a record linkage framework combined with text feature extraction and machine learning techniques. The main challenges were low data quality, lack of common record identifiers, and a limited number of other attributes shared by both data sources. Matching based solely on an exact comparison of authors' names does not solve the records linking problem because many Chinese authors share the same full name. Moreover, the English spelling of Chinese names is not standardized in the analyzed data. Three ideas on how to extend attribute sets and improve record linkage quality were proposed: (1) fuzzy matching of names, (2) comparison of abstracts of patents and articles, (3) comparison of scientists' main research areas calculated using all metadata available. The presented solution was evaluated in terms of matching quality and complexity on ≈250,000 record pairs linked by human experts. The results of numerical experiments show that the proposed strategies increase the quality of record linkage compared to typical solutions.

**Keywords:** probabilistic record linkage; fuzzy string matching; text features extraction; supervised learning; DBpedia; All Science Journal Classification (ASJC)

## 1. Introduction

Increasing amounts of collected data require the development of new effective methods for data integration, understood as the process of combining data from different sources into a unified view. Shanghai Science & Technology Talents Development Center maintain two separated databases: the Scopus database from Elsevier, containing metadata about scientific journal publications, and the Chinese Patents Database from the National Intellectual Property Administration, People's Republic of China. Integration of these databases simplifies the systems searching for experts, saves time, and reduces errors.

Data integration consists of three tasks [1]: schema matching—identifying database tables and attributes from separate data sources, record linkage—finding and linking individual records that refer to the same real-world entity, and data fusion—merging records. Human experts usually perform schema matching, but algorithms could support the most time-consuming tasks: record linkage and data fusion. This article proposes and evaluates a new solution to record linkage in the patent inventors database and scientists database.

Methods of record linkage belong to two groups: deterministic and probabilistic. Deterministic approaches link records based on exact matches between individual iden-

tifiers of two records being compared. In [2] the authors analyzed the performance of several identifiers used in deterministic record linkage. The performance of deterministic algorithms on different datasets was validated in [3,4]. The comparison of deterministic and public domain software applications was conducted in [5].

Probabilistic record linkage methods are mainly based on the Fellegi–Sunter framework [6]. Extensions include adding approximate string matching [7] or methods to reduce problem complexity [8,9]. More recent probabilistic approaches depict the record linkage problem as a binary classification problem or a clustering problem. It has been recognized [10] that the algorithm given by Fellegi and Sunter is equivalent to the Naive Bayes classifier. Other classification techniques have also been evaluated, including single-layer perceptrons [11], decision trees [12] and Support Vector Machines [13]. Record linkage as clustering was evaluated [14], using either iterative or hierarchical clustering [15,16] or graph-based techniques [17,18]. Such unsupervised learning methods are reported to give high quality linkage results, but are often impractical when used with large datasets due to their high computational requirements.

The problem of record linkage is applied mostly in the health sector [19–21], but also in national censuses [22], national security [23], bibliographic databases [24–26] and online shopping [27].

The presented algorithm links patent and scholar records, such that the scholar is the same person as one of the patent's inventors, as depicted in Figure 1.

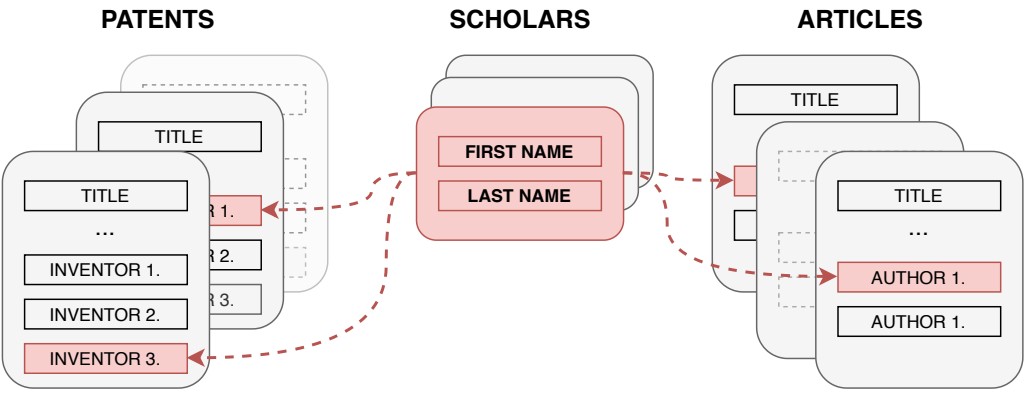

**Figure 1.** Linking patent inventors and authors of scientific articles.

The records cannot be linked using simple SQL commands because patent inventors are identified only by their names and there are no other attributes available, such addresses, birth dates, residential areas, or names and addresses of organizations. Linking records using only names is not straightforward because the way the names are stored in both databases varies. Moreover, the majority of records describe authors of Chinese origin with short and simple names. Thus, multiple authors share the same name [28]. Affiliations could not be used because the patent database does not store them.

As a baseline, we consider a simplified record linkage pipeline representing a linkage procedure performed by a human annotator without any additional knowledge about the records being linked. The baseline algorithm joins patent inventors and paper authors that have precisely the same name. All names are standardized to a common notation before joining.

To improve the quality of record linkage we propose a new algorithm that uses three strategies that involve the generation of new attributes and new methods of attribute comparison, namely: (1) fuzzy matching of names, (2) comparison of abstracts of patents and articles and (3) comparison of subject areas of patent inventors and authors of articles.

The rest of this paper is structured as follows. Section 2 contains descriptions of all record linkage steps and explanation of the algorithms and similarity functions utilized.

Section 3 provides an overview of the evaluation protocol, experiments and their results. Finally, Section 4 contains conclusions and plans for future work.

## 2. Record Linkage Algorithm

Our algorithm links patents and journal articles connected with the same scientist. Several issues make this problem challenging. Firstly, the only attributes shared between two databases are the names of scholars and patent inventors. Secondly, names are not unique and are stored and written differently, and they contain misspellings, initials, given names or family names missing, and given names and family names which are are swapped. Finally, different people can share the same name—especially Chinese authors [28]. For that reason, we built an algorithm that uses fuzzy similarities between names, compares abstracts of patents and papers, and compares subject areas (disciplines/domains) of patent inventors and authors of papers.

An indexing step reduces the number of candidate record pairs compared in detail. Indexing discards pairs that are unlikely to be true matches (i.e., it is unlikely that they refer to the same real-world entities). Without indexing, the linkage of two databases with $m$ and $n$ records, respectively, would produce $m \times n$ candidate pairs that have to be compared in detail.

In our approach, we use a combination of both standard blocking and an inverted index-based sorted neighborhood applied to English and Chinese names of scientists. Blocking [6] inserts all records that have the same value of selected attributes into the same block. The number of blocks created is equal to the number of unique values that appear in both databases. In sorted neighborhood indexing [29] matched databases are sorted according to one or more attribute values, called sorting key(s). A sliding window of fixed size (greater than one) is moved over the sorted database and candidate record pairs are generated only from the records within a current window.

All candidate pairs generated in the indexing step are subject to detailed comparisons to determine their similarity. Paired records are compared using several attributes selected from all the attributes available in the databases/tables that are linked. We use attributes depicted in Section 2.1. The results of comparisons, in the form of numerical similarity, are stored in vectors. Such comparison vectors created for each candidate record pair are inputs to classifiers depicted in Section 2.2, which decide whether a given pair is a match or a non-match.

### 2.1. Generation of Features for Comparison

In our analysis, three types of attributes are used: personal names (in English and Chinese) represented as strings, text features, extracted from abstracts and represented as numerical vectors, and All Science Journal Classification (ASJC) codes, selected directly from the database in the case of papers or predicted from the text features in the case of patents. Comparison vectors created in this step consist of a maximum of eight features representing the similarity between the records' attributes. A list of features is shown in Table 1.

**Table 1.** Overview of features used for record linking.

| Feature | Description |
|---------|-------------|
| 1 | max. exact match of Chinese names between the patent's inventors and scholar |
| 2 | max. exact match of English names between the patent's inventors and scholar |
| 3 | max. Monge–Elkan similarity of English names between the patent's inventors and scholar |
| 4 | max. SoftTFIDF similarity of English names between the patent's inventors and scholar |
| 5 | max. Extended Jaccard similarity of English names between the patent's inventors and scholar |
| 6 | max. Cosine similarity between patent's DBpedia vector and all scholars' DBpedia vectors |
| 7 | overlap coefficient between the set of ASJC codes predicted for the patent and the ASJC code assigned to the scholar |
| 8 | max. Jaccard similarity between set of ASJC codes predicted for the patent and all sets of ASJC codes of the scholar's papers |

2.1.1. The Similarity of Scientist Names, Features 1–5

Our algorithm in preliminary step completes, cleanss and normalize the English names. Chinese names were used as they are, because assessment of their quality and pre-processing requires domain knowledge.

We use and evaluate three fuzzy similarity functions specialized for multi-word comparison: Monge–Elkan, SoftTFIDF, and extended Jaccard. All three functions process the name of an author and one or more names of patent inventors. Then, we use the maximal values over all author–inventor combinations as features #3, #4, and #5 in the feature vector depicted in Table 1. For each of them, Jaro–Winkler was used as the supporting similarity function. The Jaro–Winkler similarity function is depicted in Equation (1), for two strings $s_1$ and $s_2$ counts the number of matching characters $m$, that are the same and are not further from each other than half the length of the longer string. The number of transpositions $t$, which is half the number of matching, but different sequence order characters. $l$ is the length of the matching prefix, $p$ is the prefix scale parameter.

$$sim_{jaro-winkler}(s_1, s_2) = sim_{jaro}(s_1, s_2) + lp(1 - sim_{jaro}(s_1, s_2)), \tag{1}$$

where

$$sim_{jaro}(s_1, s_2) = 1/3(m/|s_1| + m/|s_2| + (m - t)/m)). $$

The Monge–Elkan similarity function [30] extracts tokens (single words) from the two input strings and then uses a support similarity function, $sim'$, to find the most similar pairs of tokens in the sets (or bags) of tokens, as depicted in Equation (2), where $A$ and $B$ are sets of tokens extracted from $s_1$ and $s_2$, respectively.

$$sim_{monge\_elkan}(s_1, s_2) = 1/|A| \sum_{i=1}^{|A|} \max_{j=1}^{|B|} sim'(A_i, B_j) \tag{2}$$

The SoftTFIDF similarity function [31] additionally computes TF-IDF weights for the tokenized strings. This function is depicted in Equation (3), where $close(\theta, A, B)$ denote the set of all tokens $a_i \in A$ such that there is some token $b_j \in B$ such that $sim'(a_i, b_j) >= \theta$, for $0 < \theta < 1$. For $a_i \in close(\theta, A, B)$ let $S(a_i, B) = max_{b_j \in B} sim'(a_i, b_j)$. $V(i, X)$ is the TF-IDF weight of the token $i$ in the token-set $X$.

$$sim_{softtfidf}(s_1, s_2) = \sum_{u \in close(\theta, A, B)} V(u, A) \cdot V(u, B) \cdot S(u, B) \tag{3}$$

An extended Jaccard measure [32] generalizes the basic Jaccard index, which in turn calculates the similarity of the two sets, by lifting the restriction that two tokens have to be identical to be included in the set of overlapping tokens. Extension is made by utilization of a token similarity function $sim'$. The intersection of $A$ and $B$, that allows partial similarity of the tokens, $T$, is defined by Equation (4), where $\theta$ is the similarity threshold, such that $0 < \theta < 1$.

$$T = \{(a_i, b_j) | a_i \in A \wedge b_j \in B : sim'(a_i, b_j) \geq \theta\} \tag{4}$$

The extended Jaccard score is depicted in Equation (5), where $U_A$ and $U_B$ denote sets of tokens from $A$ and $B$, respectively, that are not in $T$.

$$sim_{ext\_jaccard}(s_1, s_2) = |T|/(|A \cup B|) = |T|/(|T| + |U_A| + |U_B|) \tag{5}$$

2.1.2. Title and Abstract Comparison, Feature 6

Due to the limited number of attributes shared between records in the database, we decided to search for similarities in titles and abstracts of both papers and patents, as neither full-text documents nor keywords were available. For that reason we have to transform the textual descriptions into numerical representations depicted below.

Abstracts contain not only meaningful phrases but also a lot of irrelevant words. To eliminate noise, we propose an algorithm depicted in Figure 2, We use phrases that can be mapped to the knowledge base DBpedia, which structures content stored in Wikipedia. A four-step algorithm processed a list of DBpedia entries found in text: spotting surface form sub-strings in the text that may be entity mentions, selecting candidate DBpedia resources for those surface forms, choosing the most likely candidates and filtering.

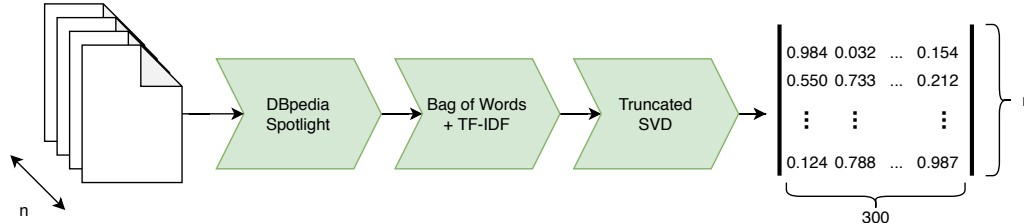

**Figure 2.** Generation of numerical representations of titles and abstracts.

We found that some of the DBpedia resources were irrelevant. Because of this, we define and apply a set of filters aiming to remove all items that point to objects, events and concepts usually not covered by scientific papers. After filtration, there were 266, 361 unique DBpedia resources extracted from the papers. These resources form a vocabulary. The resources found within patent titles and abstracts must be present in the vocabulary to be considered during comparison.

We select only the relevant phrases from paper or patent, and they were converted into a numerical representation using the Bag of Words modeling, followed by TF-IDF weighting. After transformation, we represent each paper or patent as a fixed-sized vector of real numbers. We reduce the high dimensionality of this vector (266,361) using the truncated singular value decomposition (tSVD) to 300, as suggested in [33].

The resultant vectors of length 300 created for each paper and patent were used as features in the record linkage pipeline and compared with each other using the Cosine similarity function, depicted in Equation (6), where $D_i$ and $D_J$ are concatenation of the titles and the abstracts of the *i*-th patent and the *j*-th paper, respectively, $d_i = [w_{i,1}, w_{i,2}, ..., w_{i,n}]$ and $d_j = [w_{j,1}, w_{j,2}, ..., w_{j,n}]$ are their vector representations obtained after the DBpedia resource extraction, TF-IDF weighting and reduction in dimensionality to *n* dimensions, $W_i$ and $W_j$ are $L^2$ norms of vectors $d_i$ and $d_j$.

$$sim_{cosine}(D_i, D_j) = 1/(W_i W_j) \sum_{t=1}^{n} w_{i,t} \cdot w_{j,t} \qquad (6)$$

The Cosine similarity scores are computed between the patent vector and each paper vector related to the scholar; then the maximum of these scores is selected.

### 2.1.3. Area of Interests as ASJC Codes, Features 7 and 8

The ASJC codes describe the subject areas of journals and, as a consequence, subject areas of papers published in those journals. Based on that, research areas of authors of papers can be determined, taking the most common ASJC codes assigned to his/her publications.

ASJC codes were not available for patents, and we deduced them from patent abstracts. For this task, we leveraged multi-label classification models—supervised learning models dedicated to problems where there are one or more labels per sample. We use multi-label classification models. Every journal article becomes a training sample; this means a vector with text features extracted from papers labeled with multiple ASJC codes. We apply models trained on text features extracted from papers to text features extracted from patents to determine ASJC for patents. The algorithm is depicted in Figure 3.

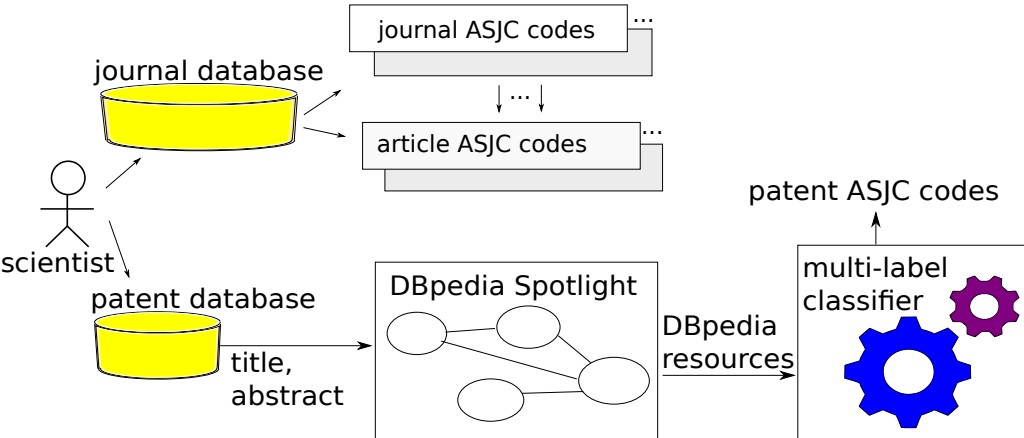

**Figure 3.** Determining ASJC codes for papers and patents.

We assumed that abstracts of both kind of documents were similar, and that sets of DBpedia resources extracted from both types of documents were overlapping. Indeed, 86.4% of DBpedia resources extracted from patents were also extracted from papers.

We implemented three multi-label classification algorithms: Binary relevance (BR), label powerset (LP), and multi-label back-propagation (BP-MLL).

BR is a method where one binary classifier is trained for each label. In BR, labels are treated independently, which is the main drawback of this method. Given an unseen instance, BR predicts a set of labels being a union of all positive results of base classifiers. An important advantage of BR is its ability to generalize beyond available label combinations. It means that BR can predict a combination of labels that were not present in training datasets. Any binary classifier might be used as a base model in BR, here a decision tree classifier (DT) was used.

LP in turn, converts each unique label combination into a separate class and applies a single multi-class classifier. The advantage of this approach is that it estimates label dependencies with only one classifier. A profound drawback of LP is that it can predict only the label combinations present in the training set. It is also prone to underfitting when the label space is large. Here, we use Gaussian naive Bayes (GNB), but any multi-class classifier can be used with LP.

BP-MLL [34] is a modification of a classical back-propagation algorithm used in the training of neural network. BP-MLL uses a novel error function capturing the characteristics of multi-label learning. It gives higher ranks to labels that belong to an instance than to those not belonging to that instance. Threshold is applied to convert achieved continuous model outputs into discrete predictions.

A set of ASJC codes predicted for a patent is first compared with a single ASJC code assigned to a scholar in the database in one of the columns of the table with authors, then it is compared with sets of ASJC codes assigned to papers authored by the scholar. As a result, we calculate two scores related to ASJC codes. The first score is just a binary indicator, whether or not an ASJC code assigned to a famous author belongs to the set of ASJC codes predicted for the patent. The second score is the maximum Jaccard similarity between ASJC codes for a patent and each set of codes assigned to papers of a scholar.

### 2.2. Classification

During the classification step, comparison vectors (Table 1) created for each candidate record pair are classified into matches or non-matches. We evaluate three types of classifiers: a simple threshold-based classifier, logistic regression, and a neural network.

A threshold-based classifier (TC) sums up partial similarity scores in each comparison vector into a single real number. Then, with the aid of one similarity threshold, it classifies all record pairs with summed similarity scores above/below the threshold value into matches/non-matches. The second classifier is cost-sensitive logistic regression (LR).

The third classifier, the multilayer perceptron (MLP), had a single hidden layer and we tuned its hyperparameters using the Hyperband [35] optimization algorithm.

### 2.3. Implementation Note

The dataset was provided as a MySQL database containing 250,000 patents, 1,100,000 patent inventor names (stored as comma separated list in each patent record), 12,000,000 journal articles and 10,000,000 authors of those articles. A total of 99% of patents and articles were published between 1990 and 2019. We extracted the data once on 15 August 2020, and the volume was 65 GB. Text stored in the databases was mostly bilingual—Chinese and English, for the purpose of this work English data were used where possible, falling back on Chinese if English translations were missing.

The portion of the provided data that was manually linked consisted of more than 200,000 patents and over 500,000 of authors. These data were used as ground-truth data for supervised learning models, depicted in Section 3.

Our system found 26,617 new links between the patent inventor and the author of the articles. Assuming a similar manual annotation speed as achieved before, our approach has saved 28 person-months of work. Of course, human experts could check algorithm results, but this process is much easier than finding connections in a large dataset.

We implemented a record linkage pipeline in the Python programming language to facilitate fast prototyping, including data transformations and visualizations. We used Python ver. 3.8.5. The indexing step is delivered by the Python Record Linkage Toolkit [36] ver. 0.14 available under the BSD-3-Clause license. We extended the comparison step of record linkage from this toolkit with fuzzy similarity measures for strings and overlap coefficient for sets from the *py_stringmatching* package ver. 0.4.1, Jaccard similarity measure from *textdistance* package ver. 4.2.0 and Cosine similarity measure from *Scikit-learn* library [37] ver. 0.23.2.

The data pre-processing step was implemented outside of the toolkit API, with the help of the Pandas [38] ver. 1.1.1 and unidecode ver. 1.0.23 libraries.

DBpedia resources were extracted using the DBpedia Spotlight [39] tool. To extract DBpedia resources from titles and abstracts of papers and patents, a local instance of DBpedia Spotlight in version 1.0 was used, with the pre-compiled English version of the language model (named: en_2+2), based on DBpedia database in version 2016-04.

Classifiers used in this project were implemented using the *Scikit-learn* library (Decision Trees, Gaussian Naive Bayes, logistic regression), the *Scikit-multilearn* framework [40] ver. 0.2.0 (Binary Relevance, Label Powerset) and *Tensorflow* [41] ver. 2.2.0 (multi-layer perceptron). Implementation of BP-MLL compatible with *Tensorflow* API was taken from an open source repository. Evaluation metrics for binary and multi-label classification (recall, precision, F-score, and other) were taken from the *Scikit-learn* library. Evaluation metrics for the indexing step were implemented from scratch.

### 3. Experiments and Discussion

#### 3.1. Evaluation of ASJC Code Prediction

Evaluation of multi-label models for ASJC code prediction was performed on article abstracts. The dataset consisted of 788,335 instances, i.e., vectors representing contents of articles, which was randomly split into training and testing subsets in a proportion of 80–20. All instances have from ground-truth labels 1 to 8 (ASJC codes).

We evaluate three multi-label models, described in Section 2.1, i.e., binary relevance with a decision tree (BR-DT), label powerset with Gaussian naive Bayes (LP-GNB) and a multilayer perceptron with adjusted back-propagation (BP-MLL). For DT a maximum tree depth of 3 was chosen, for MLP three layers, with 64, 32 and 26 neurons were used, ReLU (rectified linear unit) was used as activation in hidden layers and sigmoid in the output layer. MLP was trained for 50 epochs using an Adagrad optimizer with a learning rate set to 0.001. Outputs from the last layer were transformed into binary label indicator vectors using a constant threshold with value 0.8.

The results of multi-label classification were evaluated using a Hamming loss (HL) and example-based version of precision ($P_e$), recall ($R_e$) and $F_\beta$, depicted in Equations (7)–(10), where $n$ is the number of instances, $L$ is the set of all labels, $\Delta$ is the symmetric difference operator, $Y_i$ is the set of ground-truth labels for instance $i$, $Z_i$ is the set of predicted labels for instance $i$ and $\beta$ is a positive scale parameter.

$$HL = 1/n \sum_{i=1}^{n} |Y_i \Delta Z_i| / |L| \qquad (7)$$

$$P_e = 1/n \sum_{i=1}^{n} |Y_i \cap Z_i| / |Z_i| \qquad (8)$$

$$R_e = 1/n \sum_{i=1}^{n} |Y_i \cap Z_i| / |Y_i| \qquad (9)$$

$$F_\beta = (1 + \beta^2) \cdot (P \cdot R) / ((\beta^2 \cdot P) + R) \qquad (10)$$

Evaluation scores are presented in Table 2. In a record linking problem high recall is preferred over high precision, because, during the final record pair classification, false positives are preferred over false negatives, therefore we report $F_2$ ($\beta = 2$ in Equation (10)). This way $R_e$ is considered twice as important as $P_e$.

**Table 2.** Evaluation scores of the models in the task of ASJC code prediction for papers.

| Model | $HL$ | $P_e$ | $R_e$ | $F_2$ |
|---|---|---|---|---|
| BR-DT | 0.268 | 0.216 | 0.817 | 0.460 |
| LP-GNB | 0.127 | 0.282 | 0.490 | 0.399 |
| BP-MLL | **0.115** | **0.395** | **0.863** | **0.654** |

The best performance was achieved by the BP-MLL model, while the worst was by the LP-GNB. Both the BR-DT and BP-MLL favored the prediction of multiple labels per instance, which is noticeable in high values of $R_e$. $P_e$ of BP-MLL is superior to the other models; hence, labels predicted by this model are more accurate. The final model was created by refitting TF-IDF, tSVD and BP-MLL on whole training data.

Finally, the classifier was used for prediction of ASJC codes using patent abstracts, previously transformed using the fitted TF-IDF and tSVD models. ASJC codes decoded from the predicted binary vectors were used directly in the comparison step of the record linkage pipeline.

### 3.2. Evaluation of Record Linkage

Experiments aiming to evaluate the developed record linkage solution were conducted on records selected from the manually labeled part of the databases, which fulfil the following criteria:

1. All patent records had non-empty names (English or Chinese) for at least one inventor;
2. All patent records had non-empty titles or abstracts;
3. All scholar records had non-empty names (English or Chinese);
4. No more links can be created between records in this dataset;
5. All scholar records were linked to at least one scientific paper and this paper had non-empty title or abstract;
6. Each patent record is linked only with scholar records fulfilling 5.

We develop these criteria after a preliminary analysis of the database. We examined only the manually labeled part of the dataset. Assuming manual labels are ground truth, we have to formulate criterion 4. Among this fragment, we spotted linked records with empty values for attributes of interest: personal names, titles, and abstracts. For the baseline solution to work, we added criteria 1–3. It allowed comparison of the baseline with our

solution in terms of fuzzy name matching. Criteria 4–5 were added to assure that the other two components of our solution—comparison of abstracts and subject areas—could be tested.

This dataset consisted of records with complete values for most attributes used during record linkage (names, text features, ASJC codes). The record pairs were split into training and testing subsets using a proportion of 80:20, numbers of patents in training and testing subsets were equal to 13,502 and 4459, respectively, while the number of scholars was 16,054 in both subsets.

### 3.2.1. Evaluation of the Matching Complexity

We evaluated matching complexity using the following metrics: reduction ratio, pair completeness and pair quality. The reduction ratio (RR) provides information about how many candidate record pairs were generated by an indexing technique, compared to all possible record pairs. Pair completeness (PC) is the number of true matching record pairs that were generated, divided by the total number of true matching pairs in the full comparison space. Pair quality (PQ) is the number of candidate record pairs that correspond to true matches that were generated by an indexing technique, divided by the total number of candidate record pairs that were generated. It corresponds to precision.

In this study two indexing techniques in five different settings were compared:

1. B-EN (raw): blocking applied to raw English names;
2. B-EN: blocking applied to normalized English names;
3. B-CN: blocking applied to raw Chinese names;
4. B-CN + SN-EN: combination of B-CN and sorted neighborhood (SN) applied to normalized English names;
5. B-EN + SN-CN: combination of B-EN and sorted neighborhood applied to raw Chinese names.

The last two methods are hybrid techniques that produce a union of the candidate pairs created by each component indexing method. Additionally, the presented techniques were compared to a baseline solution, where no indexing is applied and candidates are created by taking a Cartesian product of the two datasets.

Statistics for each indexing technique calculated on a training subset are presented in Table 3. The fewest candidates were created by B-EN (raw), but PC for this method was only less than 30%. Normalization of English names improved statistics of B-EN three times when compared to B-EN (raw). B-CN outperformed B-EN in terms of PC and PQ and hybrid methods outperformed simpler techniques. A comparison of the hybrid methods on different window sizes is depicted in Figure 4.

**Table 3.** Statistics for each tested variant of indexing measured on a training subset of the reduced dataset. The best values of each statistic are in bold. For the last two methods their best variants (window = 3) are reported.

| Indexing Method | True Links | Candidates | RR | PC | PQ |
|---|---|---|---|---|---|
| Cartesian product | 13,769 | 216,761,108 | - | - | - |
| B-EN (raw) | 4084 | **6912** | **1.000** | 0.297 | 0.591 |
| B-EN | 12,607 | 45,621 | 1.000 | 0.916 | 0.276 |
| B-CN | 12,744 | 16,919 | 1.000 | 0.926 | **0.753** |
| B-CN + SN-EN | 13,695 | 114,523 | 0.999 | 0.995 | 0.120 |
| B-EN + SN-CN | **13,704** | 65,327 | 1.000 | **0.995** | 0.210 |

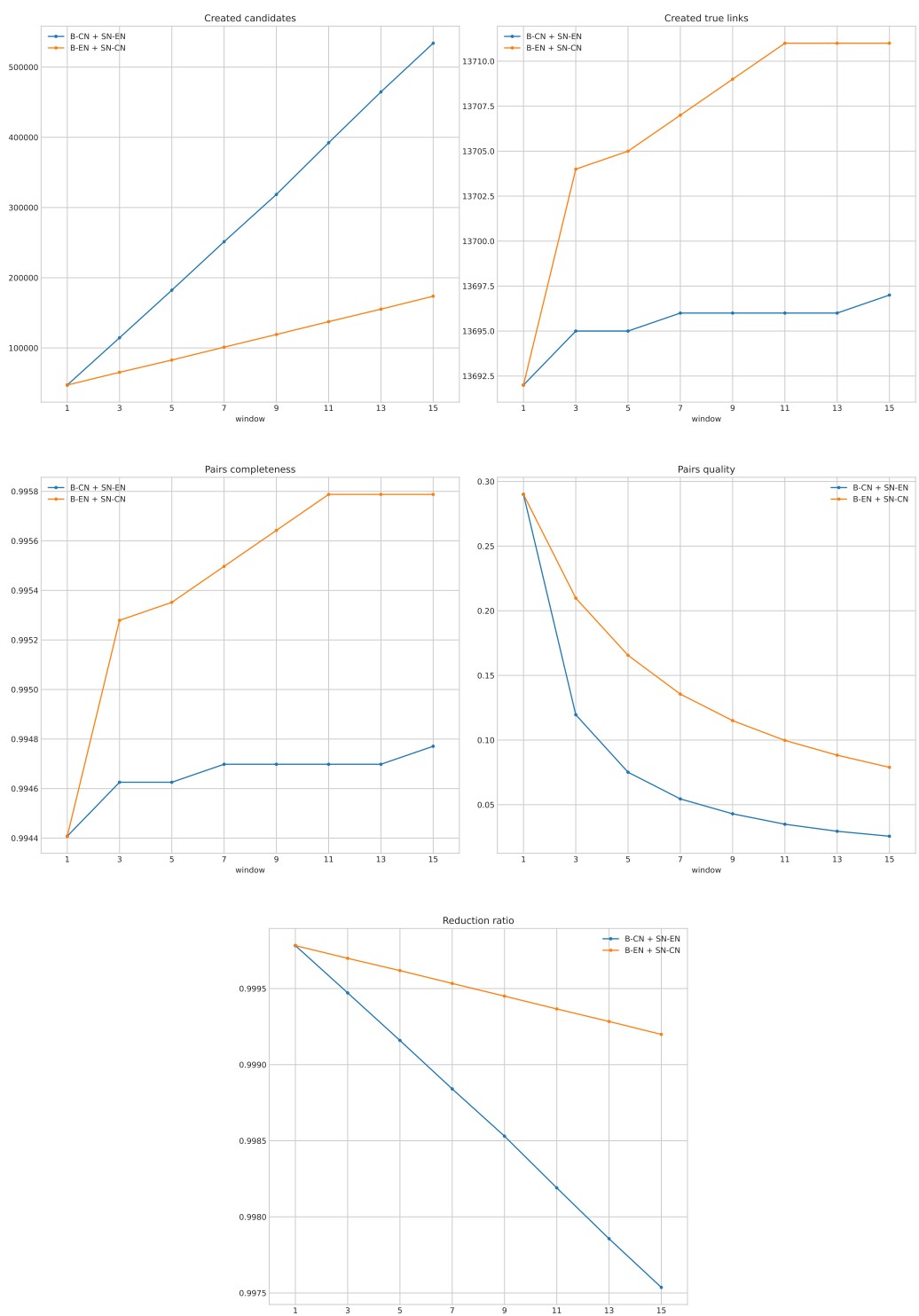

**Figure 4.** Comparison of two variants of hybrid indexing.

B-EN + SN-CN applied to the reduced dataset outperformed B-CN + SN-EN for all window sizes. A window size equal to 3 was selected as the best, because the next window (5) provided a relatively small improvement in terms of PC, but a significant increase in the number of candidates. Statistics for the selected indexing configuration measured on the testing subset are depicted in Table 4.

**Table 4.** Statistics for baseline and best variant of indexing measured on a testing subset.

| Indexing Method | True Links | Candidates | RR | PC | PQ |
|---|---|---|---|---|---|
| Cartesian product | 4540 | 71,584,786 | - | - | - |
| B-EN + SN-CN | 4510 | 26,617 | 1.000 | 0.993 | 0.169 |

### 3.2.2. Evaluation of Matching Quality

We measure the quality of record linkage using metrics utilized in the evaluation of binary classification: precision, recall and F-measure, because the final part of record linkage is the classification of previously generated candidate record pairs into matches and non-matches. We do not provide accuracy, specificity, and the false positive rate, due to high class imbalance, which is present among record pair candidates.

Three different types of classifiers, described in Section 2.2, in multiple configurations, were evaluated. Four baselines based on TC were defined. These baselines refer to the most simple methods of linking records where names of patent inventors and authors of articles are the only attributes considered and a record pair is classified as a *true-match* if both records agree on Chinese names (Baseline 1.), English names (Baseline 2.), names in either language (Baseline 3.), or finally names in both languages (Baseline 4.). Other models are trained and evaluated using different subsets of features, as depicted in Table 5.

**Table 5.** Overview of feature subsets (FS) used in experiments. Bullet depicts that the feature is in the set.

| FS＼Feat. | 1. | 2. | 3. | 4. | 5. | 6. | 7. | 8. |
|---|---|---|---|---|---|---|---|---|
| X1 | • | | | | | | | |
| X2 | | • | | | | | | |
| X3 | • | • | | | | | | |
| A | • | | • | | | | | |
| B | • | | | • | | | | |
| C | • | | | | • | | | |
| D | • | • | | | | • | | |
| E | • | • | | | | | • | |
| F | • | • | | | | | | • |
| G | • | | | • | | • | | |
| H | • | | | • | | | • | |
| I | • | | | • | | | | • |
| J | • | | | • | | • | • | |
| K | • | | | • | | • | | • |
| L | • | • | | • | | • | • | • |
| M | • | • | • | • | • | • | • | • |

During experiments 16 different feature subsets (FSs), marked with symbols X1, X2, X3 and the letters A through M, were tested. Subsets X1, X2 and X3 were used only by the baselines. FSs marked A to C used an exact match of Chinese names and different fuzzy similarities of English names. Evaluation on these FSs aimed to test the impact of fuzzy name matching on record linkage results. FS marked D extends base features with Feature 6, thus testing whether comparison of patents and papers by their content improves the results. Consequently, FSs marked E and F test how adding ASJC similarities

changes the results. FSs marked G to L contain other configurations of features that were interesting for evaluation. Finally, set M contains all the created features.

Linear correlations between features in training subsets are depicted in Figure 5. Features 1, 2, and 6 to 8 were weakly or moderately correlated with each other, with maximum Pearson's Correlation Coefficient (PCC) of 0.4 between features 1 and 6. As expected, Features 2–5 were highly correlated to each other, because they all represent similarities between the same attributes—English names of authors. Feature 4, containing a SoftTFIDF score, was arbitrarily selected out of Features 3–5 and used in the described FSs G to L. TC and LR were evaluated on each FS marked A to M and MLP was evaluated only on M, as this model was expected to have worse performance on other FSs and required separate hyperparameter tuning on each FS.

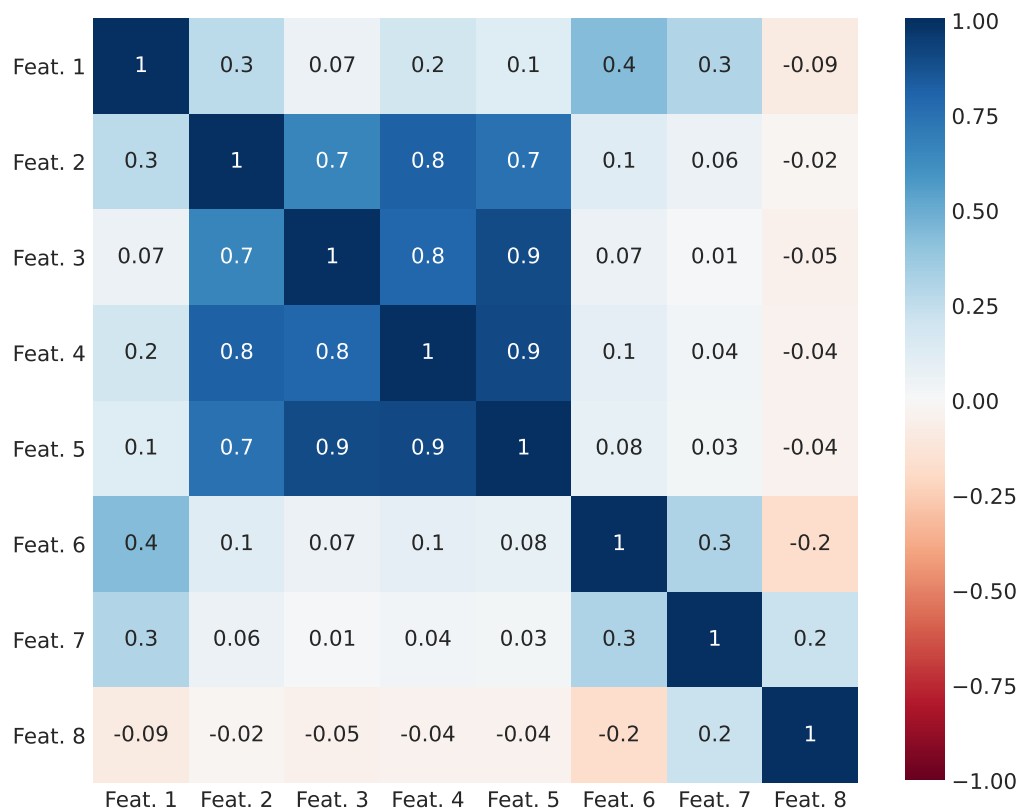

**Figure 5.** Heatmap presenting linear correlation (values of PPC) between features in the training subset.

The only parameters that were tuned for TC and LR were values of thresholds applied to the model output. In the case of TC, a threshold was applied to the sum of features and, in the case of LR, to the predicted probability of a true match. Tuning was performed using training subsets of each datasets. The performance of each model under different thresholds was compared using the precision-recall curve, as depicted in Figure 6. Each operation point on this curve corresponds to different threshold values. Thresholds with the best value of $F_1$ for each model were used during the final evaluation, where evaluation scores were calculated for both training and testing subsets of data. Finally all classifiers were ranked according to $F_1$ scores measured on testing data. In addition to precision, recall and $F_1$ scores, models were compared using a receiver operating characteristic (ROC) curve, the area under the ROC curve (ROC AUC) or Average Precision (AP), as depicted in Figure 6.

MLP required additional tuning of its hyperparameters, performed using the validation subset, created after splitting training data into two subsets in a 90:10 proportion. The split was performed in a stratified manner, to ensure that class proportion in both subsets is nearly the same. MLP consisted of two dense layers with a non-linear activation function and an additional dropout applied after the first layer. Training was performed

using the Adam optimizer. The number of units in the output layer was fixed and equal to 2 and the activation function used on this layer was softmax. This way the model produced two real-valued outputs, that refer to probabilities of two classes: match and non-match. The optimal number of units and activation function for the first layer, dropout rate and finally value of learning rate were selected during hyperparameter optimization, performed using the Hyperband algorithm [35], available in the keras-tuner library [42]. The best values of hyperparameters selected after optimization are as follows: number of units: 112, activation function: Leaky ReLU ($\alpha = 0.1$), dropout rate 0.2 and learning rate 0.01.

### 3.2.3. Evaluation of Classification

Evaluation scores for each tested combination of FS and classifier and the best threshold found, measured on training and testing subsets and sorted by ascending $F_1$ scores on testing subsets, are presented in Table 6. ROC and precision-recall curves for the best variants of TC, LR and MLP, with highlighted best operating points are depicted in Figure 6.

MLP performed best, while LP trained on M and L were ranked second and third, respectively. Adding fuzzy name matching slightly improves the results of record linkage in terms of $F_1$ scores measured for LR trained on FSs A, B and C, which were 0.003 better than the best baseline—Baseline 1. LR trained on both E and F achieved $F_1$ better then the best baseline, which confirms that comparison of subject areas improves the results of the baseline. What is interesting, is that some variants of LP that were used for comparison with record attributes other than names performed worse than LP trained on name similarities alone. This includes LP trained on D. On the one hand this result indicates that using fuzzy name matching is sometimes enough, and adding more features only reduces model performance. On the other hand, when all possible features are used, model performance is the best, as shown by LR trained on M. A conclusion can be drawn that it is better to pass all features to the machine learning model and let it decide which features are important.

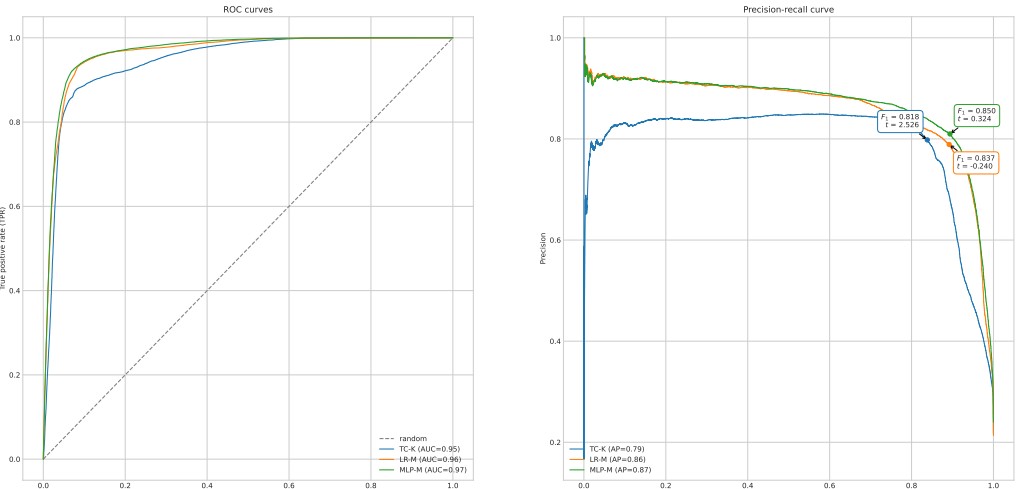

**Figure 6.** ROC and precision-recall curves for the best variants of TC, LR and MLP on the training subset of the full dataset. The best operating points on the second plot are annotated with corresponding values of $F_1$ and the threshold.

**Table 6.** Evaluation scores of the record linkage classifiers, sorted by ascending values of $F_1$ on testing subsets. The best values of each statistic are in bold. Abbreviations: B1-4 baselines, TC threshold-base classifier, LR logistic regression, MLP multi-layer perceptron, FS feature subset, AUC area under the ROC curve, AP average precision.

| Model | FS | Precision | | Recall | | $F_1$ | | AUC | AP |
|---|---|---|---|---|---|---|---|---|---|
| | | Train | Test | Train | Test | Train | Test | Train | Train |
| B2. | X2 | 0.276 | 0.271 | 0.916 | 0.913 | 0.425 | 0.418 | - | - |
| B3. | X3 | 0.290 | **0.286** | **0.994** | 0.993 | 0.449 | 0.444 | - | - |
| TC | E | 0.817 | 0.814 | 0.719 | 0.716 | 0.765 | 0.762 | 0.919 | 0.705 |
| TC | M | **0.824** | 0.821 | 0.715 | 0.710 | 0.766 | 0.762 | 0.891 | 0.737 |
| TC | L | **0.824** | 0.821 | 0.715 | 0.710 | 0.766 | 0.762 | 0.914 | 0.746 |
| TC | H | 0.816 | 0.813 | 0.742 | 0.739 | 0.777 | 0.774 | 0.920 | 0.715 |
| TC | J | 0.814 | 0.811 | 0.747 | 0.745 | 0.779 | 0.776 | 0.942 | 0.824 |
| B4. | X4 | 0.761 | 0.755 | 0.847 | 0.839 | 0.801 | 0.795 | - | - |
| TC | F | 0.758 | 0.753 | 0.851 | 0.844 | 0.802 | 0.796 | 0.912 | 0.691 |
| TC | I | 0.754 | 0.748 | 0.869 | 0.859 | 0.808 | 0.799 | 0.919 | 0.703 |
| TC | B | 0.759 | 0.754 | 0.877 | 0.869 | 0.814 | 0.807 | 0.923 | 0.704 |
| TC | C | 0.755 | 0.752 | 0.879 | 0.872 | 0.812 | 0.807 | 0.924 | 0.704 |
| TC | A | 0.755 | 0.752 | 0.879 | 0.873 | 0.812 | 0.808 | 0.924 | 0.703 |
| TC | D | 0.812 | 0.809 | 0.811 | 0.807 | 0.812 | 0.808 | 0.952 | 0.841 |
| TC | G | 0.812 | 0.808 | 0.818 | 0.813 | 0.815 | 0.811 | 0.954 | 0.846 |
| TC | K | 0.798 | 0.797 | 0.838 | 0.831 | 0.818 | 0.813 | 0.947 | 0.792 |
| LR | K | 0.747 | 0.743 | 0.929 | 0.924 | 0.828 | 0.824 | 0.964 | 0.859 |
| LR | G | 0.747 | 0.743 | 0.930 | 0.925 | 0.828 | 0.824 | 0.964 | 0.859 |
| LR | D | 0.747 | 0.743 | 0.930 | 0.926 | 0.829 | 0.824 | 0.965 | 0.859 |
| B1. | X1 | 0.753 | 0.752 | 0.926 | 0.919 | 0.831 | 0.827 | - | - |
| LR | J | 0.796 | 0.791 | 0.876 | 0.868 | 0.834 | 0.828 | 0.962 | 0.858 |
| LR | A | 0.753 | 0.752 | 0.930 | 0.925 | 0.832 | 0.830 | 0.938 | 0.727 |
| LR | B | 0.753 | 0.752 | 0.930 | 0.925 | 0.832 | 0.830 | 0.938 | 0.727 |
| LR | C | 0.753 | 0.752 | 0.930 | 0.925 | 0.832 | 0.830 | 0.938 | 0.727 |
| LR | F | 0.753 | 0.752 | 0.930 | 0.925 | 0.832 | 0.830 | 0.943 | 0.745 |
| LR | I | 0.753 | 0.752 | 0.930 | 0.925 | 0.832 | 0.830 | 0.943 | 0.745 |
| LR | E | 0.753 | 0.752 | 0.930 | 0.925 | 0.832 | 0.830 | 0.950 | 0.775 |
| LR | H | 0.753 | 0.752 | 0.930 | 0.925 | 0.832 | 0.830 | 0.948 | 0.776 |
| LR | L | 0.790 | 0.788 | 0.879 | 0.878 | 0.832 | 0.831 | 0.964 | 0.859 |
| LR | M | 0.789 | 0.787 | 0.892 | 0.887 | 0.837 | 0.834 | 0.965 | 0.861 |
| MLP | M | 0.810 | 0.810 | 0.894 | 0.885 | **0.850** | **0.846** | **0.968** | **0.868** |

## 4. Conclusions

The evaluation results showed that fuzzy name matching, comparison of abstracts of patents and papers and comparison of ASJC codes improved matching quality when compared to the baseline solution that links records only by name attributes. Experiments conducted on different feature subsets indicate that the performance of the threshold-based classifier strongly depends on the feature subsets used. On the other hand, classification methods based on machine learning perform best on complete feature sets.

Our classification system uses ASJC codes at the article level, but the ASJC codes are assigned at the journal level. This simplification may lead to some bias, especially for non-mainstream articles and multi-disciplinary works. Usage of title, keywords and abstract only partially solve this problem. The multi-label classification helps to reduce false negatives and helps here. We checked dozens of articles and did not find errors by assuming that journal ASJC codes accurately depict each article.

Moreover, multi-label classification can properly handle multi-disciplinary and interdisciplinary researchers. Our algorithm assigns a set of scientific domains to every researcher. The domains, ASJC codes, could belong to similar or different scientific areas. The risk of improper results for scientists with multi-/interdisciplinary studies exists when such a scientist is the author of papers/patents from different disciplines A and B and she/he has no paper/patent classified for both A and B. In such a case, the algorithm returns two scientists with the same name: the first scientist publishes in discipline A and the second scientist publishes in discipline B. If databases have an article or a patent for this scientist in both disciplines, this article/patent links two clusters creating one. We manually checked our databases and found no such cases.

ASJC codes were not available for patents, but patents, in turn, are categorized using codes from the International Patent Classification (IPC) system. ASJC and IPC systems are entirely disjointed, and mapping between them does not exist. We examined both sets of codes and we attempted to map them, first manually, then using machine learning techniques. Neither method gave satisfactory results.

The presented solution automates the process of record linkage in the Shanghai Talents Center. Integration of the patents database and scientific articles database improves systems' abilities to search for experts and gives a better description of scientists and helps to detect cases where authors first publish a substantial part of their invention as an article in a scientific journal and then want to patent it.

In a production environment, our classifier may produce both false positives and false negatives. To mitigate the former, we proposed to collect record pairs marked by the algorithm as true matches and present them to domain experts who will make a final decision. Due to the presence of a large number of record pairs, we determined another threshold. Results produced by the final classifier with confidence between the lower and upper threshold are presented to annotators and results with confidence above the upper threshold are accepted without manual approval. Fighting with a high number of false negatives is more complicated. Moreover, our proposal is lowering values of the threshold applied to classifier outcomes; however, it leads to a higher number of record pairs that are presented to domain experts.

Further work includes implementation of the proposed solution in production systems and evaluation on similar datasets, for example on the Polish Patents Database.

**Author Contributions:** Conceptualization, R.N. and W.F.; methodology, R.N.; software, W.F.; validation, R.N., W.F. and X.T.; resources and data curation, W.F., X.T., Z.Z., X.C. and X.L.; writing, R.N. and W.F.; supervision, J.Z. and Y.Z.; funding acquisition, R.N. and Y.Z. All authors have read and agreed to the published version of the manuscript.

**Funding:** This research was funded by Shanghai-Warsaw Scientific Joint Lab and Institute of Computer Science Statutory Funds.

**Institutional Review Board Statement:** Not applicable.

**Informed Consent Statement:** Not applicable.

**Data Availability Statement:** Not applicable.

**Conflicts of Interest:** The authors declare no conflict of interest.

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
