# Peer review of "Record Linkage of Chinese Patent Inventors and Authors of Scientific Articles"

_applsci, doi:10.3390/app11188417_

Round 1
Reviewer 1 Report
Dear Authors,
As one of the reviewers, I read your submission and found it a well-written research article. congratulations. My only concern is your problem section, becuase I couldn't easily understood what is your research problem until I advanvced through the paper. So, I suggest you to present a clear and explicit research problem to make it easy for your target reader.
I wrote this comment on your introduction, but because I have no more comments, instead of sending you the file, attach it here:
"I think it is necessary to ease the understanding of the readers by an explicitly research problem.
I think the research problem is to integrate two databases that are referred in the first paragraph, and then the possible methods are explained. I suggest the authors to make it clearer."
Best of luck
Author Response
Dear Reviewer,
We are glad that in your opinion our article is well-written, thank you.
We addressed your comment, "I think it is necessary to ease the understanding of the readers by an explicitly research problem. I think the research problem is to integrate two databases that are referred in the first paragraph, and then the possible methods are explained. I suggest the authors to make it clearer."
modifying text in 2 places. We added a sentence in the abstract (lines 1-3)
"We present an algorithm to find corresponding authors of patents and scientific articles. The authors are given as records in Scopus and the Chinese Patents Database. This issue is known as the record linkage problem, defined as finding and linking individual records from separate databases that refer to the same real-world entity."
And we added text in the introduction (lines 26-28).
"Data integration consists of three tasks [9]: schema matching -- identifying database tables and attributes from separate data sources, record linkage -- finding and linking individual records that refer to the same real-world entity, and data fusion -- merging records. Human experts usually perform schema matching, but algorithms could support the most time-consuming tasks: record linkage and data fusion. This article proposes and evaluates a new solution to record linkage in patent inventors database and scientists database."
Sincerely yours,
Robert Nowak,
for the authors
Reviewer 2 Report
This paper presents an innovative way to record details of patents and scientific articles. More details can be given to show the seriousness of the “record linkage” problem to highlight the significance of the research. Why are the machine learning techniques considered appropriate to be chosen for this research? Is there any validation on the results of the numerical experiments to illustrate the proposed strategies increase the quality of record linkage compared to typical solutions? What do the ‘typical solutions’ refer to (in Abstract)? How do the authors justify that the methods of record linkage should belong to deterministic and probabilistic groups but not others? While Fig. 1 presents the algorithm between patent and scholar records, similar figure can be used to show the set of ASJC codes assigned to a scholar in the database. How to carry out the three fuzzy similarity functions specialized for multi-word comparison? More elaborations are required to show how the authors make the decision to compare papers and patents in the database by their titles and abstracts to make the research more rigorous. How can the algorithm proposed in Fig. 2 deal with the problem of irrelevant words in Abstracts? How to ensure the accuracy of the process? More explanations are required for the choice of multi-label classification models when ASJC codes are not available for patents. How to ensure that well over half (i.e., 86.4%) of DBpedia resources extracted from patents are also extracted from papers? Why should a decision tree classifier be used in the research? How do supervised learning models compare with the machine learning models? How does the use of Python language facilitate the achievements of the research goals? Are there any implications from the best performance achieved by the BP-MLL model? One important part of the research is the evaluation of record linkage. How are the criteria set to evaluate the developed record linkage solution? Is normalization necessary for the raw Chinese names? In conclusions, the authors assume that journal ASJC codes accurately depict each article. How valid is this assumption? The comments as mentioned above should be addressed to ensure that the paper is of the required quality standard to be considered for publication by Applied Sciences.
Author Response
Dear Reviewer,
thank you for your constructive comments. We addressed all of them. The detailed answer is given below.
Issue 1: This paper presents an innovative way to record details of patents and scientific articles.
More details can be given to show the seriousness of the “record linkage” problem to highlight the significance of the research.
ANSWER:
To highlight a record linkage problem we added the definition in the abstract (lines 1-3) and we added description in the introduction (lines 26-28).
Issue 2: Why are the machine learning techniques considered appropriate to be chosen for this research?
ANSWER:
We do research on machine learning, which is why we choose the presented method for this research. Hopefully, we achieve promising results, so we consider our methods are appropriate to the presented problem.
Issue 3: Is there any validation on the results of the numerical experiments to illustrate the proposed strategies increase the quality of record linkage compared to typical solutions?
What do the ‘typical solutions’ refer to (in Abstract)?
ANSWER:
In the presented article, we compare several algorithms using numerical experiments. Every algorithm is faster than the currently used record linking process by a human expert, who decides if records should or should not be connected.
On the other hand, we use the human expert decision on the testing dataset as ground-truth so that machine learning algorithm quality could be at most as good as a human expert quality.
Typical solutions (in the abstract) refer to linkage methods based solely on exact authors' names comparison, like a SQL join clause.
Those methods' results fall behind our solution. It produces either too few record pairs (even minor inconsistencies in spelling of the names make comparisons fail) or too many record pairs (a lot of false positives, because many Chinese authors share the same name and further distinction is impossible without additional information).
Issue 4: How do the authors justify that the methods of record linkage should belong to deterministic and probabilistic groups but not others?
ANSWER:
When record identifiers are used in both data sources, they are compared directly to select matching records - this approach is deterministic. Otherwise, when there are no shared identifiers, other record features (not identifiers) are analyzed. In this case new features could be created, comparisons are not limited to the exact matching but used together with similarity functions. The given set of record pairs, called training dataset or ground-truth dataset, is analyzed to discover conditions or rules to match two records. This approach we call a probabilistic approach. All automated record linkage methods fall into those two categories. A separate category is the manual approach, but it requires human labor, time, and expert knowledge.
Issue 5: While Fig. 1 presents the algorithm between patent and scholar records, similar figure can be used to show the set of ASJC codes assigned to a scholar in the database.
ANSWER:
Yes, we agree. We added new figure to present ASJC codes prediction algorithm.
Fig. 3, titled "Determining ASJC codes for papers and patents", page 6.
Issue 6: How to carry out the three fuzzy similarity functions specialized for multi-word comparison?
ANSWER:
Thank you for this comment. It is not clear in the text, thereore we change lines 118-124.
Currently the text is:
"We use and evaluate three fuzzy similarity functions specialized for multi-word comparison: Monge-Elkan, SoftTFIDF, and extended Jaccard.
All three functions process the name of an author and one or more names of patent inventors.
Then, we use the maximal values over all author-inventor combinations as features \#3, \#4, and \#5 in the feature vector depicted in Tab.~1. For each of them, Jaro-Winkler was used as the supporting similarity function."
Issue 7: More elaborations are required to show how the authors make the decision to compare papers and patents in the database by their titles and abstracts to make the research more rigorous.
ANSWER: We agree with the reviewer and we change lines 136-139. Currently, the text is as follow:
"Due to the limited number of attributes shared between records in the database, we decided to search for similarities in titles and abstracts of both papers and patents, as neither full-text documents nor keywords were available. For that reason we have to transform the textual descriptions into numerical representations depicted below."
Issue 8: How can the algorithm proposed in Fig. 2 deal with the problem of irrelevant words in Abstracts?
ANSWER: The algorithm uses the DBpedia Spotlight - a tool that analyzes text and searches for mentions of DBpedia entities in it. The irrelevant words are omitted, because only meaningful phrases and entities are included in DBpedia. Unfortunately, we found that some of the extracted entities were still irrelevant to the abstracts' contents. Thus we filtered out DBpedia entities referring to objects, persons, events, and concepts usually not covered by scientific papers. Finally we have a filtered list of DBpedia entities for each abstract that are all relevant. We provide this information in lines 146-150.
Issue 9: More explanations are required for the choice of multi-label classification models when ASJC codes are not available for patents.
ANSWER: Thank you for this comment, it is not clear in the text. We change it, currently (lines 171-177) it is as below.
"
ASJC codes were not available for patents, and we deducted them from patent abstracts. For this task, we leveraged multi-label classification models - supervised learning models dedicated to problems where there is one or more label per sample.
We use multi-label classification models.
Every journal article becomes a training sample, it means a vector with text features extracted from papers labeled with multiple ASJC codes. We apply models trained on text features extracted from papers to text features extracted from patents to determine ASJC for patents."
Issue 10: How to ensure that well over half (i.e., 86.4%) of DBpedia resources extracted from patents are also extracted from papers?
ANSWER: It is good comment. In our opinion it is impossible to ensure a specific percentage of overlapping DBpedia resources. The more articles in the paper set, the longer is a list of extracted DBpedia resources, and the more significant are chances that they overlap with resources extracted from patents. It is important to note that ASJC code similarities are not the only features used in our solution. In case there are no overlapping resources, record linkage is still possible with fuzzy name similarities.
Issue 11: Why should a decision tree classifier be used in the research?
ANSWER: Thank you for examining our algorithm very carefully.
Our research has evaluated decision trees and other binary classifiers like logistic regression, Naive Bayes. Finally, we decide to use a decision tree classifier for binary relevance as it yielded decent results and took less time to complete. In the presented paper, we should determine what part of the research to include, and we decided to focus on comparing different multi-label algorithms. From that perspective, selecting a binary classifier for binary relevance or a multi-class classifier for label powerset is hyper-parameter tuning.
Issue 12: How do supervised learning models compare with the machine learning models?
We use only supervised models, calling them for simplicity machine learning models. We think it is common practice.
Issue 13: How does the use of Python language facilitate the achievements of the research goals?
ANSWER:
For our research, we decided to use Python language,
as it facilitates fast prototyping - including data transformations, data visualizations, and availability of libraries like Python Record Linkage Toolkit, Scikit-Learn, Scikit-multilearn, Tensorflow, etc. We modify the paragraph in section 2.3 (lines 235-242), currently it is
"We implemented a record linkage pipeline in the Python programming language to facilitate fast prototyping, including data transformations and visualizations. We used Python ver.~3.8.5.
The indexing step is delivered by the \emph{Python Record Linkage Toolkit}~\citep{de_bruin_j_2019_3559043}
ver.~0.14 available under the BSD-3-Clause license.
We extended the comparison step of record linkage from this toolkit with fuzzy similarity measures for strings
and overlap coefficient for sets from the \emph{py\_stringmatching}
package ver.~0.4.1,
Jaccard similarity measure from \emph{textdistance} package ver.~4.2.0 and Cosine similarity measure from
\emph{Scikit-learn} library~\citep{scikit-learn} ver. 0.23.2."
Issue 14: Are there any implications from the best performance achieved by the BP-MLL model?
ANSWER: In our opinion multi-label problems are usually complex, thus they require complex models to solve them.
In ASJC codes prediction task, models that replace complex multi-label problem with simpler problems (BR, LP) achieved inferior results to the BP-MLL, a model designed specifically for multi-label tasks.
Issue 15: One important part of the research is the evaluation of record linkage. How are the criteria set to evaluate the developed record linkage solution?
ANSWER: Thank you for this comment. We completly agree that the criteria should be given. We added paragraph in section 3.2 lines 300-307.
"We develop these criteria after a preliminary analysis of the database.
We examined only the manually labeled part of the dataset. Assuming manual labels are ground truth, we have to formulate criterion 4.
Among this fragment, we spotted linked records with empty values
for attributes of interest: personal names, titles, and abstracts.
For the baseline solution to work, we added criteria 1-3. It allowed comparing the baseline with our solution in terms of fuzzy name matching. Criteria 4-5. were added to assure that the other two components of our solution - comparison of abstracts and subject areas - could be tested."
Issue 15: Is normalization necessary for the raw Chinese names?
ANSWER:
Normalization of Chinese names is not necessary,
although it could potentially improve the results.
We omitted this normalization due to insufficient knowledge
of the authors in terms of Chinese transcription systems,
spelling rules and lack of trustworthy automated solutions.
Issue 16: In conclusions, the authors assume that journal ASJC codes accurately depict each article. How valid is this assumption?
ANSWER:
The article published in journal is checked by Editors and Reviewers in term of relevance,
therefore we assume that ASJC codes are manually assigned. It is simplification and we mentioned it in Conclusion, lines 431-436.
"Our classification system use ASJC codes at the article level, but the ASJC codes are assigned at the journal level. This simplification may lead to some bias, especially for non-mainstream articles and multi-disciplinary works. Usage of title, keywords and abstract only partially solve this problem. The multi-label classification helps to reduce false negatives and helps here.
We checked dozens of articles and did not find errors by assuming that journal ASJC codes accurately depict each article."
Thank you for a lot of constructive comments.
Sincerely yours,
Robert Nowak,
for the authors